# Macro Model for Discrete-Time Sigma-Delta Modulators

**Kye-Shin Lee**

Department of Electrical and Computer Engineering, The University of Akron, Akron, OH 44325, USA; klee3@uakron.edu; Tel.: +1-330-972-2996

**Abstract:** This work presents a macro model for discrete-time sigma-delta modulators, which can significantly reduce the simulation time compared to transistor level circuits. The proposed macro model is realized by effectively combining active and passive ideal circuit components with Verilog-A modules. As such, since the macro model is a true representation of the actual transistor level circuit, a moderately good accuracy can be obtained. In addition, the proposed macro model includes the major amplifier, comparator, and switch-capacitor non-idealities of the sigma-delta modulator such as amplifier DC gain, GBW, slewrate, comparator bandwidth, hysteresis, parasitic capacitance, and switch-on resistance. The results show the simulation time of the proposed macro model sigma-delta modulator is only 6.43% of the transistor level circuit with comparable accuracy. As a result, the proposed macro model can facilitate the circuit design and leverage non-ideality analysis of discrete-time sigma-delta modulators. As a practical design example, a second order discrete-time sigma-delta modulator with a five-level quantizer is realized using the propose macro model for GSM and WCDMA applications.

**Keywords:** macro model; non-idealities; sigma-delta modulator; signal-to-noise-distortion ratio (SNDR); simulation time

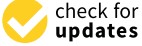



## 1. Introduction

Nowadays, the most important requirement for analog-to-digital converters (ADCs) are low-power and high-speed operation. The two main factors driving low-power operations are VLSI technology scaling toward the deep sub-micron feature size and increasing the demand for battery-operated portable electronic devices such as RF handsets to prevent excess heating and enable longer battery operation time. In addition, high-speed operation is demanding for emerging telecommunication applications to handle wider frequency bands. Sigma-delta modulators are widely used for low-bandwidth and high-resolution applications including voice, audio, instrumentation, and data acquisition, owing to its oversampling and noise shaping properties [1–3]. Since the concept of the sigma-delta modulator was first proposed in 1962 [4], sigma-delta modulators have evolved due to the increased demands on higher bandwidth and resolution. In particular, multi-bit quantizers [5], higher order modulators [6], multi-stage noise shaping (MASH) structures [7], and time-interleaved architectures [8] have been realized to extend the bandwidth and resolution. However, due to the oversampling nature of sigma-delta modulators, even with the above-mentioned improvements, the high sampling rate is the main obstacle that leads to long simulation time.

Macro and behavioral models can be very useful in the design stage of the sigma-delta ADCs, since it can predict the performance of the ADC, which provides a guideline for the actual circuit implementation. Moreover, choosing the optimized ADC architecture and estimating the requirements for each building block is a very time-consuming procedure using only transistor level design and simulation. In addition, the SPICE Modeling Lab. no longer provides strong and weak model parameters for process technologies beyond 65 nm. In this case, with transistor level simulations, the only solution is to use worst case statistical corners obtained from each building block using Analog Circuit Studio (ACS) or

Monte Carlo (MC) analysis. However, this requires running the ADC top level simulations for each set of worst-case corners to obtain good simulation coverage [9].

The behavioral or macro modeling approach can overcome this problem, since behavioral and macro models do not directly use statistical transistor models. Instead, the circuit parameters such as DC gain, gain bandwidth product (GBW), slewrate, and offset obtained from each building block under the worst case corners can be combined into a single simulation for the purpose of estimating the performance of the sigma-delta modulator. So far, a number of behavioral models for discrete-time sigma-delta modulators have been proposed [10–12]. The authors of [10] present a time-domain behavioral model realized using Matlab and Simulink, where major non-idealities of the sigma-delta modulator including kT/C noise, jitter, amplifier finite DC gain, GBW, and slewrate are modeled. In [11], there is also a Simulink-based time-domain behavioral model that incorporates amplifier, comparator, and switch non-idealities. However, unlike other approaches, this work models amplifier DC gain non-linearity, which can affect the performance, especially for sigma-delta modulators using single-bit quantizers that have large amplifier output swings. In [12], two behavioral models realized with Matlab, Simulink, and VHLD-AMS are presented [13]. The first model is based on a switch-capacitor (SC) integrator transient response including the effects of amplifier non-idealities and the dynamic capacitive loading effect on the settling time. The second model is based on symbolic node admittance matrix representation of the system, where non-idealities such as jitter, thermal noise, and DAC mismatch are included in the sigma-delta modulator model. Although behavioral models show fast simulation times compared to transistor level circuits, since behavioral models are mathematical representations of the actual circuit, there are limitations for modeling the true circuit behavior including fully differential operation, response to errors, and non-idealities. Therefore, a modeling approach for sigma-delta modulators that can accurately represent the true circuit behavior is in high demand. However, there is not much work on sigma-delta modulator macro models that can represent the actual circuit operation and the effect of non-idealities with considerable simulation time reduction compared to transistor level sigma-delta modulators.

In this work, to overcome the limitations of existing behavioral models, a macro model for discrete-time sigma-delta modulators is proposed. Compared to the behavioral models [10–13], the proposed macro model replaces the actual circuit blocks of the sigma-delta modulator including the amplifier, comparator, and SC integrator with ideal circuit components and Verilog-A modules, instead of modeling the sigma-delta modulator with mathematical expressions. Thus, the actual circuit topology, circuit operation, and response to errors and non-idealities are preserved in the macro model. Moreover, since macro models can be easily combined with transistor level circuits, they can be used throughout the entire design stage, with significant simulation time reduction compared to transistor level circuits, whereas the usage of behavioral models are mainly limited to the initial design stage. Eventually, the proposed macro model will contribute to non-ideality analysis and design time reduction for discrete-time sigma-delta modulators. The remaining portion of the paper is organized as follows. The proposed discrete-time sigma-delta modulator macro model is described in Section 2, the macro model simulation results with major circuit non-idealities are shown in Section 3, and the conclusions are provided in Section 4.

## 2. Proposed Macro Model

### 2.1. Sigma-Delta Modulator Non-Idealities

Figure 1 shows the block diagram of a second order discrete-time sigma-delta modulator that is widely used for various applications due to its high power/area efficiency and good stability [14,15]. The circuit level sigma-delta modulator can be divided into analog and digital blocks. The analog block consists of the SC integrator (with the amplifier), comparator, and feedback DAC, whereas the digital block contains the clock generator, logic gates, and dynamic element matching (DEM) circuit.

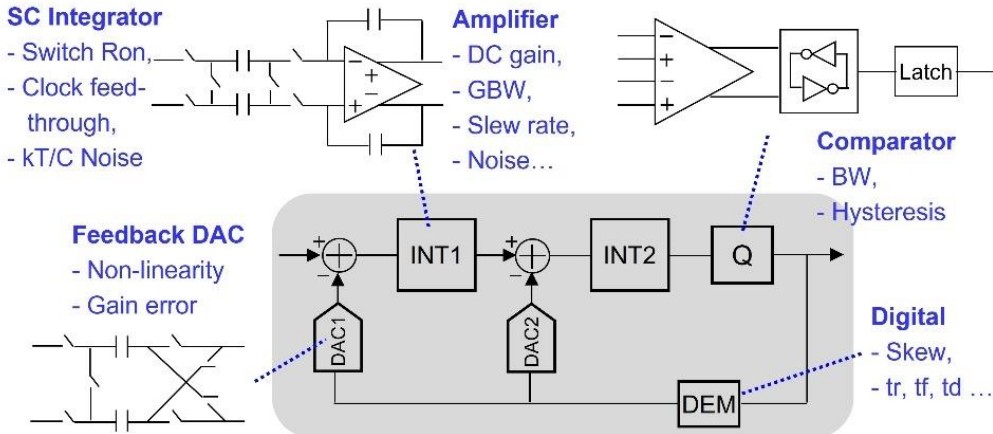

**Figure 1.** Block diagram of a second order discrete-time sigma-delta modulator.

The major error sources that are in the analog block includes the amplifier finite DC gain, GBW, slewrate, offset, SC integrator kT/C noise, switch-on resistance, channel charge injection, clock feedthrough, capacitor mismatch, comparator bandwidth, hysteresis, and feedback DAC non-linearity. As such, the proposed macro model incorporates the major analog block circuit non-idealities that are based on quantitative analysis of amplifier, comparator, and SC integrator non-idealities presented in [10–13]. However, among the various non-idealities, the emphasis will be put on the error sources that the macro model can better describe compared to the behavioral models.

### 2.2. Amplifier Macro Model

Figure 2 shows the concept of the amplifier macro model that includes the piecewise linear (PWL) trans-conductance $g_m$, output resistance $R_o$, and output capacitance $C_o$. The PWL $g_m$ can represent the slewing, settling region, and the transition between the slewing and settling region, where $+I_o$ and $-I_o$ are the maximum current driven by the PWL $g_m$ voltage controlled current source (VCCS) [12]. In this case, the DC gain $A_o$, gain bandwidth product GBW, and slewrate $SR_+$ and $SR_-$ are provided as:

$$A_o = -g_m \cdot R_o \tag{1}$$

$$GBW = g_m / (2\pi C_o) \tag{2}$$

$$SR_+ = +I_o / C_o \tag{3a}$$

$$SR_- = -I_o / C_o \tag{3b}$$

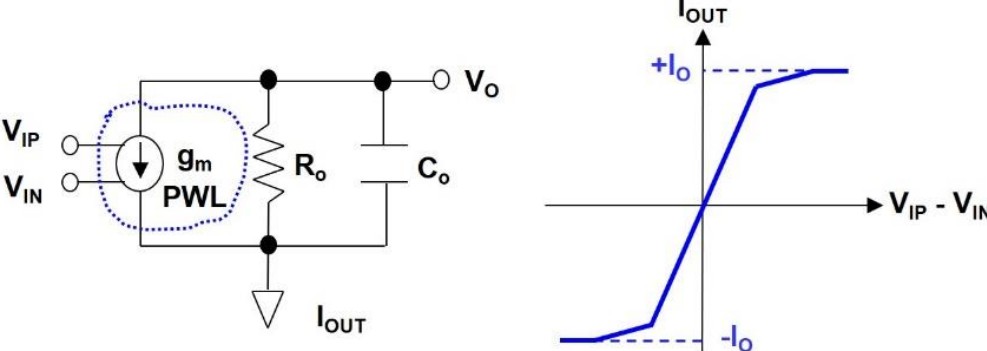

**Figure 2.** Concept of amplifier macro model.

Figure 3 shows the amplifier macro model, where two sets of PWL $g_m$ ($g_{mp}$, $g_{mn}$), input/output resistance ($R_{ip}$, $R_{in}$, $R_{op}$, $R_{on}$), and input/output capacitance ($C_{ip}$, $C_{in}$, $C_{op}$, $C_{on}$) are used to model the fully differential (FD) operation [16]. As such, $g_{mp}$ and $g_{mn}$ represents the input stage that is generally realized with differential pairs [17]. In addition, the input common mode (CM) level $V_{I,CM}$ is set by using the average of the two inputs $V_{ip}$ and $V_{in}$. This is realized with the input resistance and input capacitance network, where $R_{ip}$ and $R_{in}$ are set to 1 TΩ, since the MOS amplifiers have very high input impedances and $C_{ip}$ and $C_{in}$ represent the input parasitic of the amplifier. The output CM level $V_{O,CM}$ is generally set to $V_{DD}/2$ (for single supply) or 0 V (for dual supply). For the actual amplifier, $V_{O,CM}$ is set by the common mode feedback (CMFB) circuit [18]. As a result, the amplifier macro model will work for both differential and CM inputs. Assuming $R_{op} = R_{on} = R_o$ and $C_{op} = C_{on} = C_o$, the amplifier output is provided as:

$$V_{op} - V_{on} = A(s) \cdot [V_{ip} - V_{in}] \tag{4}$$

where $A(s) = -g_m \cdot R_o / (1 + sR_oC_o)$. For CM inputs, the amplifier output will be $V_{op} = V_{on} = V_{O,CM}$.

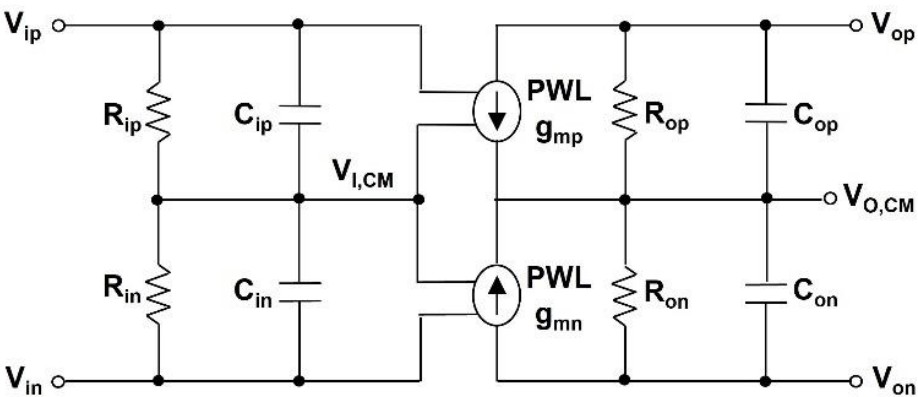

**Figure 3.** Amplifier macro model.

### 2.3. Comparator Macro Model

Figure 4 shows the concept of the comparator macro model. The comparator model is based on the latched comparator topology that consists of the pre-amp., regenerative (R) latch, and set-reset (SR) latch. Latched comparators are widely used in sigma-delta modulators due to their low power consumption and fast responses even with small input difference and good kick-back noise suppression [19]. In addition, latched comparators generally use double differential (DD) amplifiers for the pre-amp., where the input difference ($V_{ip} - V_{in}$) is again subtracted from the reference input difference ($V_{Rp} - V_{Rn}$) to generate the output. As a result, the pre-amp. output is provided as:

$$V_{op} - V_{on} = A_c \cdot [(V_{ip} - V_{in}) - (V_{Rp} - V_{Rn})] \tag{5}$$

where $A_c$ is the pre-amp. gain (usually set between 2 and 10) and the reference inputs $V_{Rp}$ and $V_{Rn}$ are generated by a resistive ladder. The pre-amp. output is further processed by the R-latch so that the output difference ($V_{op} - V_{on}$) is enlarged and the SR-latch generates the final digital output (H or L).

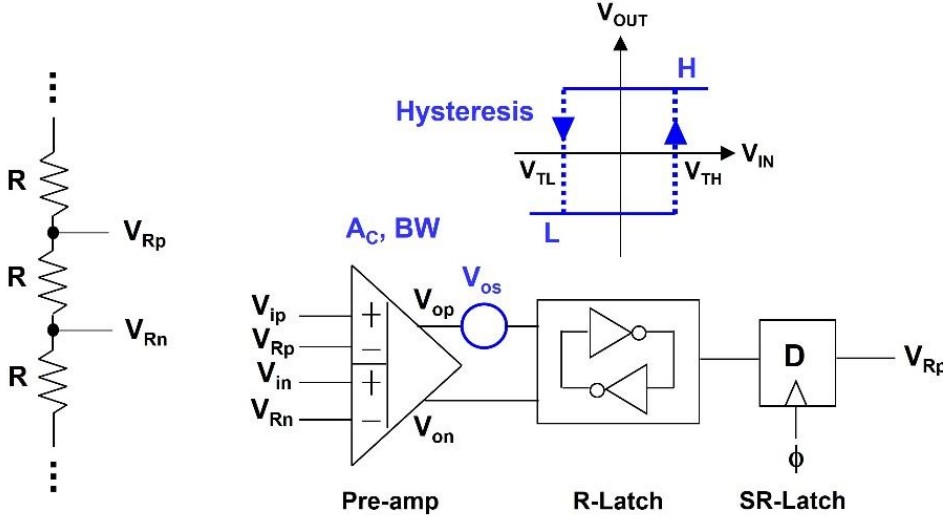

**Figure 4.** Concept of comparator macro model.

The major non-idealities of the latched comparator include the pre-amp. gain $A_c$, bandwidth BW, R-latch offset, and hysteresis. The latch hysteresis is a memory effect in the comparator that tends to cause it to stay in the previous output state, which results from the charge stored in the internal capacitors [20]. This alters the comparator threshold leading to a false decision. That is, if the previous output state is H, the actual threshold becomes lower than the nominal threshold, once the output state changes back to L and vice versa when the previous output state is L. Furthermore, the amount of hysteresis is provided as:

$$\text{Hystresis} = V_{TH} - V_{TL} \tag{6}$$

where $V_{TH}$ and $V_{TL}$ is the upper and lower threshold of the comparator, respectively.

Figure 5 shows the comparator macro model, where the DD pre-amp. is realized with voltage-controlled voltage sources (VCVS), input impedance $R_{ci}$, $C_{ci}$ and output impedance $R_{co}$, and $C_{co}$. However, unlike the amplifier, since the pre-amp. operates in the open-loop configuration, the slew rate is not critical, thus the pre-amp. is modeled with $VCVS_1$ and $VCVS_2$ instead of the PWL VCCS. The output of the pre-amp. is provided as:

$$V_{\text{pre\_amp\_p}} = A_c \cdot (V_{ip} - V_{Rp}) \tag{7a}$$

$$V_{\text{pre\_amp\_n}} = A_c \cdot (V_{in} - V_{Rn}) \tag{7b}$$

where $A_c$ is the pre-amp. gain with the bandwidth BW of $1/(2\pi R_{co} C_{co})$. Furthermore, the R-latch and SR-latch realized using $VCVS_3$ with gain $A_L$ and the latch mode time constant $\tau_L = R_L \cdot C_L$ and Verilog-A based analog latch and limiter. In this case, $A_L$ and $\tau_L$ determines the positive feedback operation of the R-latch [21], where although $R_{ic}$ and $C_{ic}$ do not directly affect the performance of $VCVS_3$, the $R_{oc}$ and $C_{oc}$, which are equivalent to the input impedance of the R-latch, determine the preamp. bandwidth. As a result, the limited pre-amp. bandwidth can generate a false input to the R-latch ($VCVS_3$), leading to a comparator error. The output of $VCVS_3$ will be latched at the rising edge of the latch clock $CLK_L$, where the limiter will scale the R-latch output to generate the final digital level output H or L. Moreover, the hysteresis $V_{TH}$ and $V_{TL}$ will be added or subtracted from the reference voltage $V_{Rp}$ or $V_{Rn}$, depending on the previous output state. This is equivalent to increasing or decreasing the comparator threshold level. That is:

$$V_{Rp} \to V_{Rp} + V_{TH} \text{ and } V_{Rn} \to V_{Rn} - V_{TL} \tag{8a}$$

$$\text{(for previous output } V_{comp} = H) \tag{8a}$$

$$V_{Rp} \rightarrow V_{Rp} - V_{TL} \text{ and } V_{Rn} \rightarrow V_{Rn} + V_{TL} \qquad (8b)$$

$$\text{(for previous output } V_{comp} \text{ L)} \qquad (8b)$$

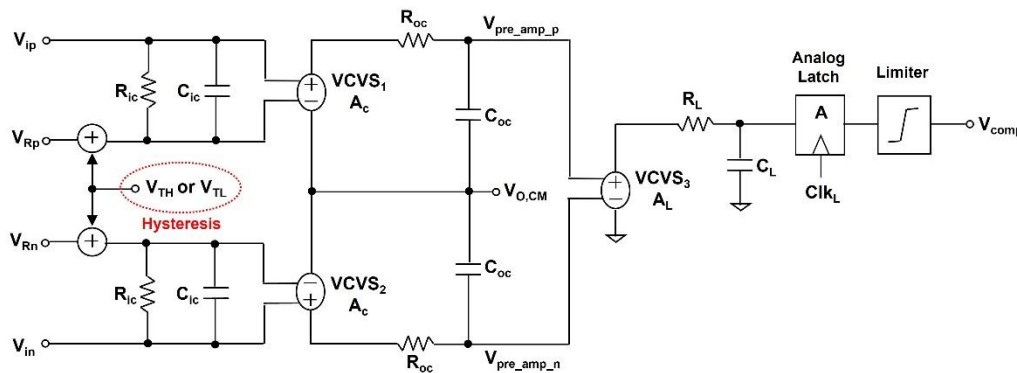

**Figure 5.** Comparator macro model.

### 2.4. SC Integrator Macro Model

Figure 6 shows the concept of the SC integrator macro model, which includes the amplifier, sampling capacitor $C_S$, feedback capacitor $C_F$, DAC capacitor $C_{DAC}$, and switches, where the switches are controlled by the two-phase non-overlapping clocks $\Phi1$ and $\Phi2$ and $V_{R+}$ and $V_{R-}$ are the reference voltages. For high order sigma-delta modulators with multiple integrators, the first integrator is the most critical block, since the input signal is directly applied to the first integrator. The non-idealities of the SC integrator are switch-on resistance $R_{on}$, channel charge injection, parasitic capacitance $C_{p1,2}$, capacitor mismatch $\Delta C_{S,F}$, and amplifier offset $V_{os}$. In particular, limited amplifier SR can cause incomplete charge transfer between $C_S$ (or $C_{DAC}$) and $C_F$. That is, the charge stored in $C_S$ during the sampling phase ($\Phi1 = H$, $\Phi2 = L$) should properly transfer to $C_F$ during the integration phase ($\Phi1 = L$, $\Phi2 = H$) through the discharging current flowing across $C_S$ and $C_F$, where the limited amplifier SR can suppress the discharging current, leading to error at the integrator output.

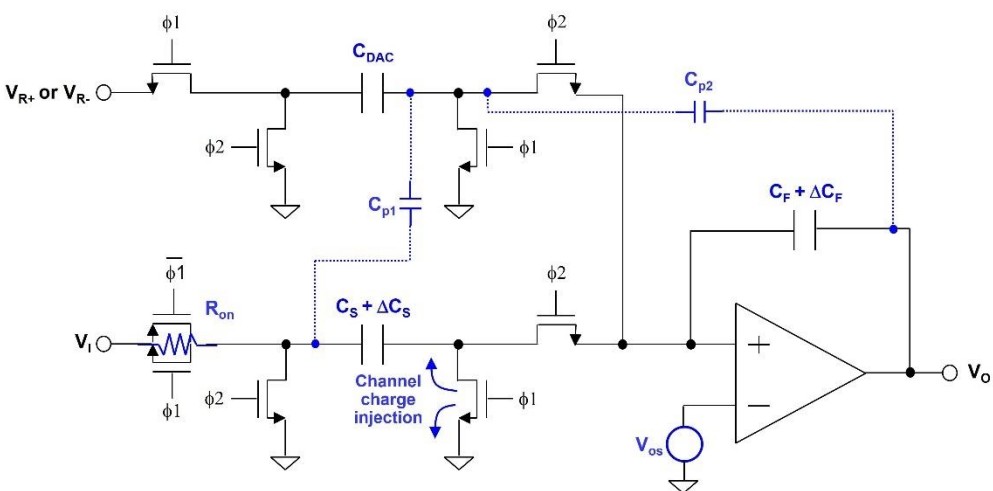

**Figure 6.** Concept of SC integrator macro model.

Figure 7 shows the SC integrator macro model that is based on the fully differential stray insensitive configuration [22]—this is the most widely used SC integrator configuration for sigma-delta modulators. The non-idealities included in the proposed macro model are switch-on resistance and parasitic capacitance, since the effect of the two non-idealities are not straight forward to model using behavioral models. For the proposed macro model, the error due to the switch-on resistance and parasitic capacitance can be accurately represented, since the original circuit topology is preserved with one to one correspondence

for the components between the actual circuit and the macro model. However, the effect of kT/C noise and the channel charge injection are not considered, since kT/C noise can be nullified by properly sizing the sampling capacitor $C_S$ and the channel charge injection will not critically degrade the modulator performance, in case the bottom plate sampling technique is used with minimum sized switches [23].

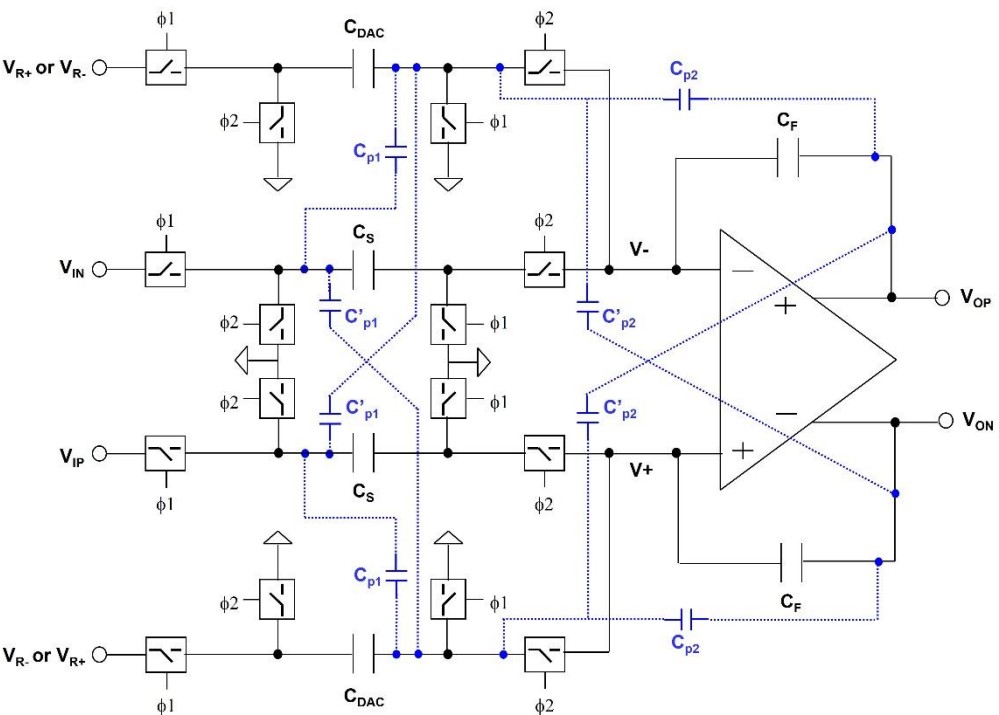

**Figure 7.** SC integrator macro model.

The switches are realized with a Verilog-A module with adjustable on/off resistance, where the on/off states are set by the control signal. That is, when the switch is on or off, the resistance between the two terminals are set to the on-resistance and the off-resistance, respectively. As a result, the on/off resistance of each switch can be set individually. The parasitic capacitance is mainly due to layout parasitic that leads to cross talk or coupling. Although the SC integrator is realized with the stray-insensitive configuration, the parasitic coupling between the input and output nodes can critically degrade the modulator performance. However, since the parasitic capacitance coupled to the ground or reference nodes will be effectively nullified by the stray insensitive configuration, these are not included in the macro model. Therefore, only the most critical parasitic that is coupled between the summer node ($v_+$, $v_-$) and the input/output nodes are included [24]. However, assuming a symmetrical layout—this is the general case for fully differential circuits, the parasitic coupling through the positive and negative input/output terminals are assumed to be identical.

## 3. Macro Model Simulation Results

### 3.1. Simulation Setup

The second order sigma-delta modulator with first and second integrator coefficients of 0.5 and 2, respectively, and a five-level quantizer (block diagram shown in Figure 1) are realized using the proposed amplifier, comparator, and SC integrator macro model. The sampling rate $f_s$ of the modulator is set to 19.2 MHz with oversampling rate (OSR) of 71 and 4.8, respectively, which enables the signal bandwidth of 135 kHz (for GSM) and 2 MHz (for WCDMA). Table 1 shows the macro model parameter default values used for the simulation. The macro model sigma-delta modulator with input frequency of 51.526 kHz is simulated with Spectre Mixed Mode Circuit Simulator$^{TM}$, where multiple simulations

were performed by varying the macro model parameter values. For each simulation, the modulator output spectrum was obtained from an 8192-point FFT, where the signal-to-noise and distortion ratio (SNDR) are measured from the output spectrum. SNDR is a widely used metric for evaluating the dynamic performance of sigma-delta modulators, since SNDR shows both the noise level and the harmonic power at the output. However, in order to verify the accuracy of the proposed macro model, the output spectrum of the macro model sigma-delta modulator is compared with the transistor level sigma-delta modulator realized using CMOS 65 nm technology with a supply voltage of 1.3 V. Figure 8 shows the output spectrum of the transistor level (based on nominal corner models) and macro model sigma-delta modulator using default parameter values shown in Table 1. The SNDR difference between the macro model and the transistor level sigma-modulator showed 0.5 dB (for GSM band) and 0.1 dB (for WCDMA band), respectively, which validates the accuracy of the proposed macro model.

**Table 1.** Sigma-delta modulator macro model parameter default values.

| Parameter [Unit] | Amplifier-1 | Amplifier-2 |
|---|---|---|
| $g_m$ [uA/V] | 200 | 100 |
| $R_o$ [MΩ] | 10 | 10 |
| $C_i$ [fF] | 50 | 25 |
| $C_o$ [fF] | 25 | 10 |
| $V_{os}$ [mV] | 1 | 2 |
| SR [V/us] | 100 | 50 |
| | **Integrator-1** | **Integrator-2** |
| $C_S$ [fF] | 250 | 200 |
| $C_F$ [fF] | 500 | 100 |
| $R_{on}$ [Ω] | 1 k | 1 k |
| $C_{p1}$ [fF] | 3 | 3 |
| $C_{p2}$ [fF] | 2 | 2 |
| $C'_{p1}$ [fF] | 1 | 1 |
| $C'_{p2}$ [fF] | 0.5 | 0.5 |
| | **Comparator** | |
| $A_c$ [V/V] | 5 | |
| $R_{oc}$ [MΩ] | 0.3 | |
| $C_{oc}$ [fF] | 10 | |
| $V_{TL}$ [mV] | −5 | |
| $V_{TH}$ [mV] | +5 | |
| $A_L$ [V/V] | 20 | |
| $R_L$ [MΩ ] | 0.2 | |
| $C_L$ [fF] | 10 | |

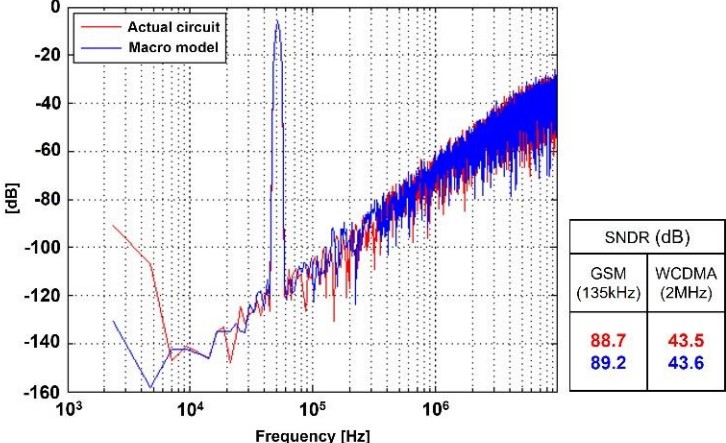

**Figure 8.** Transistor level and macro model sigma-delta modulator output spectrum.

### 3.2. Effect of Amplifier Non-Idealities

The effects of amplifier DC gain, GBW, and slewrate on the modulator performance are evaluated using the proposed macro model. The default macro model parameters shown in Table 1 will set $A_o = 70$ dB, GBW = 500 MHz, and SR = 120 V/us. However, when the value of one variable either $A_o$, GBW, or SR is changed, the remaining two variables are set to the default values. Figure 9a shows the modulator output spectrum with different $A_o$ values, where $A_o$ is varied by changing $R_o$ to keep the GBW and SR fixed to the default values. The limited $A_o$ cause integrator leakage—a certain portion of the charge stored in the feedback capacitor $C_F$ will not be added with the sampled charge, which mainly increases the modulator in-band noise level [10]. The SNDR of the modulator showed 89.2 dB (GSM band) and 43.6 dB (WCDMA band) for $A_o = 70$ dB and 62.2 dB (GSM band) and 32.4 dB (WCDMA band) for $A_o = 30$ dB. Furthermore, GBW is varied by changing $C_o$ and SR is varied by changing the slope of the PWL $g_m$. As shown in Figure 9b,c, limited GBW and SR cause improper charge transfer between the sampling and the feedback capacitor, thus leading to both in-band noise floor increments and harmonic distortion. The modulator SNDR showed 89.2 dB (GSM band) and 43.6 dB (WCDMA band) for GBW = 500 MHz but degrade to 73.5 dB (GSM band) and 36.3 dB (WCDMA band) for GBW = 20 MHz. The impact of SR on modulator SNDR showed 88.5 dB (GSM band), and43.2 dB (WCDMA band) for SR = 120 V/us and 53.9 dB (GSM band) and 40.2 dB (WCDMA band) for SR = 10 V/us.

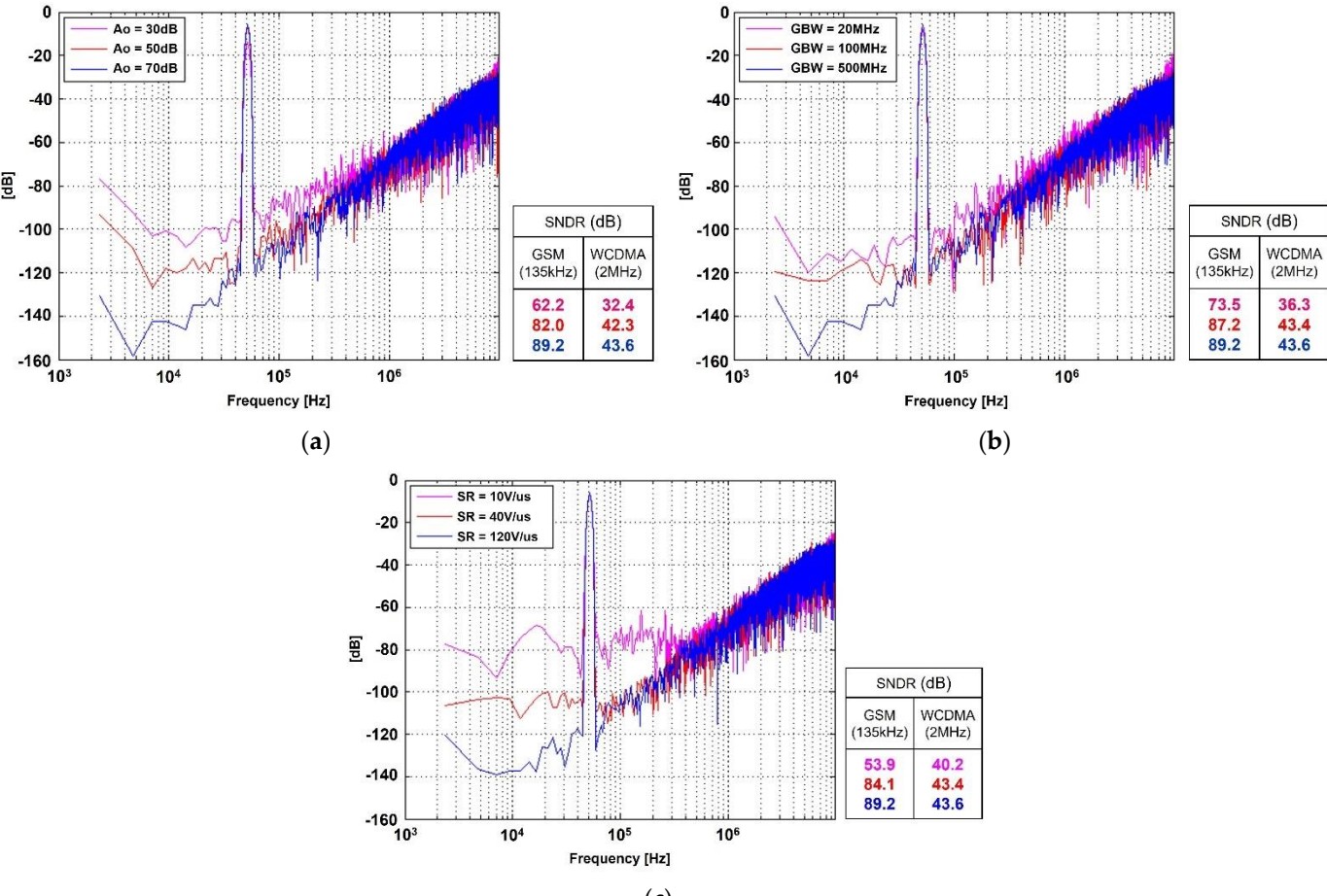

**Figure 9.** Modulator output spectrum with amplifier non-idealities. (**a**) $A_o$, (**b**) GBW, and (**c**) SR.

### 3.3. Effect of Comparator Non-Idealities

For the comparator, the default macro model parameters shown in Table 1 set the pre-amp. bandwidth to BW = 53.1 MHz, hysteresis = 10 mV, and R-latch time constant $R_L \cdot C_L$ = 2 ns. However, since the comparator offset will be noise-shaped by the loop filter, it will not critically degrade the modulator performance, thus the modulator can tolerate a large offset [25]. As a result, the most critical non-idealities that are pre-amp. bandwidth BW and hysteresis will be evaluated using the proposed macro model. Figure 10a shows the modulator output spectrum with different BW, where a limited pre-amp. bandwidth can cause comparator input amplitude dependent errors. This increases the in-band noise floor and shifts the output spectrum toward the signal band. The modulator SNDR showed 89.2 dB (GSM band) and 43.6 dB (WCDMA band) for BW of 50 MHz and 70.9 dB (GSM band) and 27.8 dB (WCDMA band) for BW of 2.5 MHz. Figure 10b shows the modulator output spectrum with different comparator hysteresis, where hysteresis can also lead to comparator input amplitude dependent errors. However, hysteresis mainly causes hormonic distortion at the modulator output. The modulator SNDR showed 88.1 dB (GSM band) and 43.1 dB (WCDMA band) for hysteresis of 10 mV and 74.3 dB (GSM band) and 39.4 dB (WCDMA band) for hysteresis of 40 mV.

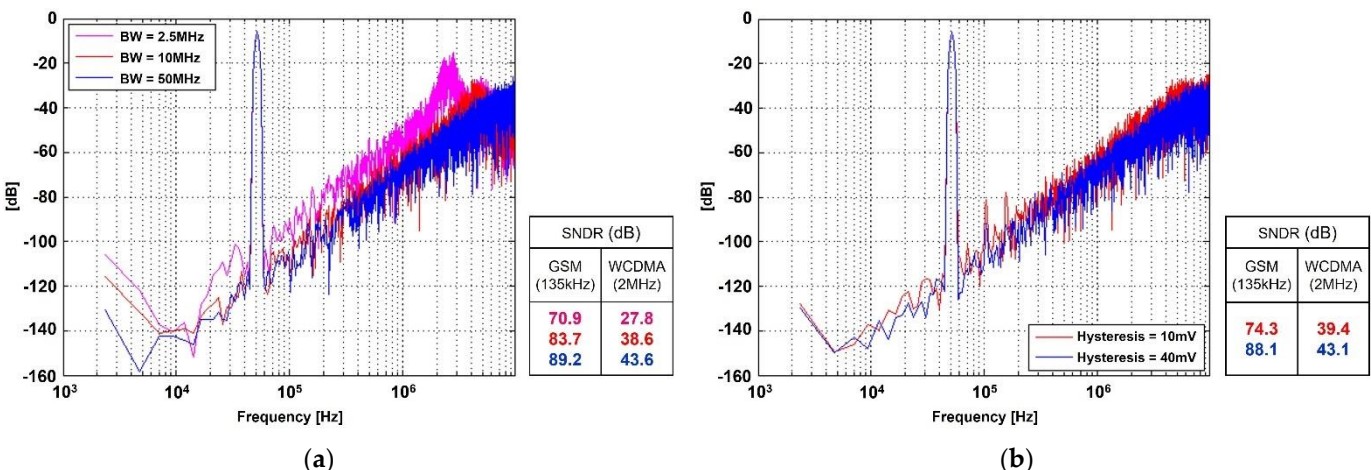

**Figure 10.** Modulator output spectrum with comparator non-ideality. (**a**) Pre-amp bandwidth and (**b**) Hysteresis.

### 3.4. Effect of SC Integrator Non-Idealities

Among the SC integrator non-idealities, the effect of switch-on resistance $R_{on}$ and parasitic capacitance are evaluated using the proposed macro model, since the other non-idealities including kT/C noise and channel charge injection can be sufficiently modeled by behavioral models. Figure 11a shows the modulator output spectrum with different $R_{on}$ values, where limited $R_{on}$ shows similar effect as the limited amplifier GBW that cause improper chare transfer, leading to signal dependent errors. As such, both the in-band noise level and the harmonics are increased at the modulator output. The modulator SNDR showed 89.1 dB (GSM band) and 43.4 dB (WCDMA band) for $R_{on}$ of 1 kΩ, but decreased to 78.9 dB (GSM band) and 43.3 dB (WCDMA band) for $R_{on}$ of 20 kΩ. Figure 11b shows and considers the modulator output spectrum with and without parasitic capacitance, the most critical parasitic coupling between the input nodes ($V_{IP}$, $V_{IN}$) and summing nodes ($V+$, $V-$) through $C_{p1}$ and $C'_{p1}$, and between the summing nodes and output nodes ($V_{OP}$, $V_{ON}$). In this case, each parasitic was set to a different value (refer to Table 1), which is the worst case [24]. As shown, the parasitic capacitance mainly leads to distortion and slightly increases the in-band noise level. However, even with more or less 1% parasitic with respect to $C_S$ and $C_F$, the modulator SNDR decreased from 89.2 dB (GSM band) and 43.6 dB (WCDMA band) to 76.3 dB (GSM band) and 43.3 dB (WCDMA band).

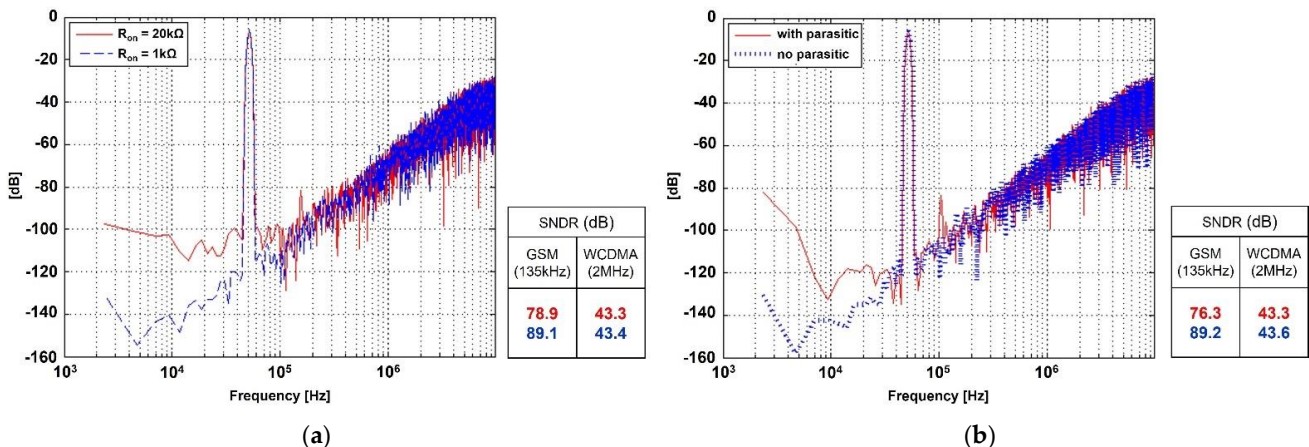

**Figure 11.** Modulator output spectrum with SC integrator non-ideality. (**a**) Switch-on resistance (**b**) Parasitic capacitance.

*3.5. Simulation Time Comparison*

The simulation time for the proposed sigma-delta modulator macro model is compared with the transistor level sigma-delta modulator circuit. In this case, to better compare the simulation time, the analog and digital blocks of the sigma-delta modulator are replaced with the transistor level circuit or the macro model. The analog block includes the amplifier, SC integrator, comparator pre-amp, R-latch, and feedback DAC and the digital block consists of the control clock generator, comparator SR-latch, and logic. For fair comparison, the transistor level and the macro model sigma-delta modulators used input signal frequency of 51.526 kHz with sampling rate of 19.2 MHz, where 8192 output samples were obtained to perform the FFT. Figure 12 shows the simulation time of each sigma-delta modulator, where the simulation time of the macro model sigma-delta modulator showed only 6.43% compared to the all transistor level sigma-delta modulator.

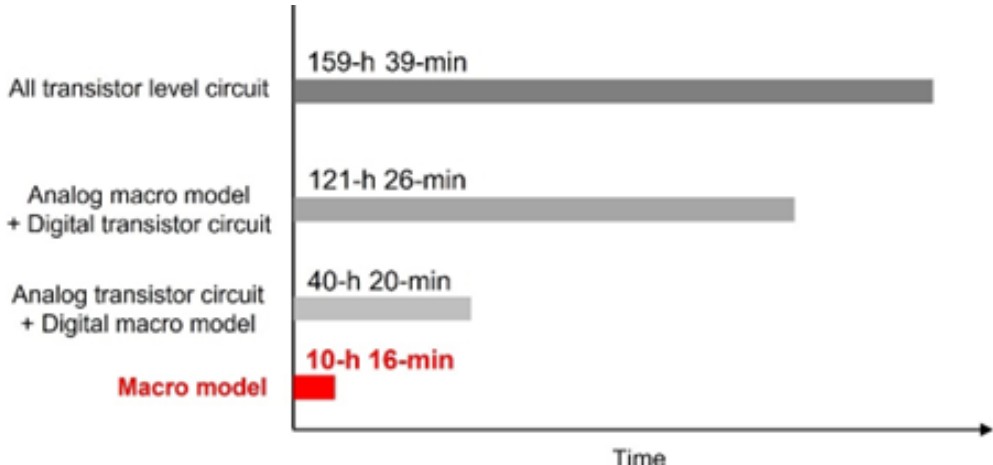

**Figure 12.** Simulation time comparison for the transistor circuit and macro model sigma-delta modulator.

## 4. Conclusions

In this work, a macro model for discrete-time sigma-delta modulators have been provided. The proposed macro model is realized by effectively combining active and passive ideal circuit components with Verilog-A modules. As such, moderately good accuracy can be obtained with reduced simulation times. The proposed macro model covers the major amplifier, comparator, and switch-capacitor non-idealities of the sigma-delta modulator such as amplifier DC gain, GBW, slewrate, comparator bandwidth, hysteresis, parasitic capacitance, and switch-on resistance. The results show the simulation time of

the proposed macro model sigma-delta modulator is only 6.43% of the transistor level circuit with comparable accuracy. As a result, the proposed macro model can facilitate the circuit design and leverage non-ideality analysis of discrete-time sigma-delta modulators. As a practical design example, a second order discrete-time sigma-delta modulator with a five-level quantizer for GSM and WCDMA applications is realized using the proposed macro model, where the effect of major non-idealities on the modulator performance has been evaluated.

**Funding:** This research received no external funding.

**Data Availability Statement:** Not applicable.

**Conflicts of Interest:** The author declares no conflict of interest.

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
