# Peer review of "Macro Model for Discrete-Time Sigma‒Delta Modulators"

_electronics, doi:10.3390/electronics11233994_

Round 1

Reviewer 1 Report

The first criticism is that the literature review is poor. Old and lumped references are presented in section 1, but an in-depth discussion is not derived after all. Sigma delta modulators have been thoroughly addressed since the 1970s, but this is still a modern research topic. What has been done so far? What are the research gaps? The contribution is not clear compared with the current state of the art.

I understand that the authors propose a model for representing sigma delta modulators. Even so, how are small differences in the components used in Fig. 5, that is, Ric, Cic, Coc, and Roc supposed to affect VSC3 in Fig. 5?

How does the slew rate of the operational amplifier in Fig. 6 influence the integrator behavior? Please, elaborate.

SR is often used in the analysis of sigma delta modulators. Why did you choose SNDR as a metric instead?

The conditions adopted in the assessment of simulation times and comparison of modulators in section 3.5 must be provided.

Last but not least, you must proofread the whole manuscript to eliminate typos, as well as ensure the correct use of verbal tenses in all sentences. Please, replace all figures with high-quality vector graphics, because all of them are blurred when zooming in.

Reviewer 2 Report

This article presents a macro model for discrete-time sigma-delta modulators, which significantly reduces the simulation time compared to transistor level circuits. The simulation results are presented. Some concerns need to be addressed, which are listed as below:

(1) It is suggested to verify the proposed method under corner cases. Monte Carlo simulation results of the method are better to be added.

(2) The novelties, significances, and technical contributions of the macro model for sigma-delta modulators are suggested to be presented.

(3) The figures are suggested to be presented in a more clear format.

(4) The significant differences with other methods and the mechanism of the proposed model are better to be highlighted.

(5) It would be better if the model implementation can be illustrated with algorithms and quantitative analysis added.
